# Eye Tracking-Based Characterization of Fixations during Reading in Children with Neurodevelopmental Disorders

**DOI:** 10.3390/brainsci14080750

**Published:** 2024-07-26

**Authors:** Carmen Bilbao, Alba Carrera, Sofia Otin, David P. Piñero

**Affiliations:** 1Department of Optometry, Hospital Quirón, 22003 Huesca, Spain; carmenbill0@gmail.com (C.B.); albass.acb@gmail.com (A.C.); 2Department of Applied Physics, University of Zaragoza, 50009 Zaragoza, Spain; sofotin@unizar.es; 3Group of Optics and Visual Perception, Department of Optics, Pharmacology and Anatomy, University of Alicante, San Vicente de Raspeig, 03690 Alicante, Spain; 4Department of Ophtalmology, Vithas Medimar International Hospital, 03016 Alicante, Spain

**Keywords:** fixation, oculomotor disfunction, eye tracker, dyslexia, attention deficit/hyperactivity disorder, developmental coordination disorder

## Abstract

An efficient mode of evaluation for eye movements is the use of objective eye tracking systems combined with subjective tests (NSUCO or DEM), which are easily applicable across all age groups and in eye care clinical settings. The objective of this study was to characterize fixations during reading in two groups: a group of children with neurodevelopmental disorders (NDDG, 24 children, age: 6–12 years) and a group of children with oculomotor anomalies but without NDD (OAG, 24 children, age: 6–12 years). The results obtained were compared with those from a control group (CG, 20 children, age: 6–12 years). Specifically, the outcomes obtained with two subjective score systems, the Northeastern State University College of Optometry’s Oculomotor (NSUCO) test and the Developmental Eye Movement (DEM) test, were compared with the objective analysis obtained through a commercially available eye tracker (Tobii Eye X, Tobii, Stockholm, Sweden). Specialized analysis software, namely Clinical Eye Tracker 2020 (Thomson Software Solutions, Welham Green, UK), was used. It was found that children with NDD had impaired oculomotor skills. A higher number of regressions, more fixations, and longer durations of fixations appear to be characteristic signs of this population group. Additionally, children with NDD took longer to complete the DEM test, as well as exhibiting more errors. The use of objective videoculographic systems for eye tracking and subjective tests like the NSUCO or DEM are good tools to assess saccadic movements, allowing the detection of oculomotor abnormalities in children with NDD.

## 1. Introduction

Oculomotor function is the ability of humans to move their eyes naturally, in a coordinated and smooth manner, while maintaining a clear, fused, and fixed image at the central point of the retina [1]. Other significant types of eye movement include vergence and the vestibulo-ocular reflex (VOR). Although this study primarily focuses on eye movements related to reading, it is worth mentioning that some researchers have linked vergence instability to dyslexia [2]. Three main types of eye movement are commonly used to characterize oculomotor skills: fixations, smooth movements, and saccades. Fixations are the voluntary ability to hold the gaze on a specific visual stimulus. However, the eyes do not remain completely immobile during fixations; they exhibit small tremors, drifts, and entirely involuntary, independent microsaccades for each eye, with amplitudes of less than 1°. These fixational eye movements are responsible for keeping the image on the fovea to prevent the obtained image from appearing blurry, as performing these small movements avoids the saturation of the photoreceptors [3,4]. Saccades are rapid eye movements that create a fixation and bring the object point to the center of the retina, being the fastest type of movement that the human body can generate [5]. In the study of eye movements, especially during reading, several technical terms are used. Regressions during reading are backward eye movements where the reader’s gaze returns to previously read text. These movements are often involuntary and can occur when the reader needs to reprocess or clarify the content. Regressions are typically associated with difficulties in comprehension or in decoding the text, reflecting the reader’s need to revisit earlier words or sentences to enhance their understanding. They are a natural part of the reading process but are more frequent in individuals with reading difficulties. According to recent research, skilled readers typically exhibit fewer than 15% of their eye movements as regressions per 100 words, indicating higher efficiency in processing text compared to individuals with reading difficulties [6,7]. Antisaccades are voluntary eye movements where an individual must suppress a reflexive saccade towards a visual stimulus and instead look in the opposite direction. This task requires significant cognitive control, involving the inhibition of the automatic response and the generation of a deliberate saccade in the opposite direction [7,8]. These different types of eye movement are crucial in the development of reading [9] and can have a negative impact on it when an oculomotor problem is present. Therefore, the evaluation of oculomotor function, specifically the analysis of fixation movements, should be considered a necessary assessment in eye examinations for children [10].

Several studies have shown that oculomotor skills can be limited or altered in different health conditions, including neurological [11] or neurodevelopmental disorders (NDD). NDD are disorders accompanied by specific learning difficulties with a pattern of specific signs and an intelligence quotient within the normal range, although they are often associated with poor school performance. These disorders encompass many difficulties, including dyslexia, attention deficit disorder with or without hyperactivity (ADHD), and developmental coordination disorder (DCD) [12]. Dyslexia is a neurobiological disorder that affects a person’s ability to read, write, and process language effectively. It is defined as the inability to develop the ability to read at an expected level, despite having normal intelligence. There is controversy about whether people with dyslexia have oculomotor deficiencies in addition to their deficiencies in the rapid processing of visual information [13]. ADHD is a neurobiological disorder characterized by difficulties in maintaining attention, impulsivity, and excessive activity that is not appropriate for the child’s developmental stage [14]. DCD is characterized by motor coordination skills that are below what is expected for the individual’s age. These children have difficulties in the acquisition and automation of motor activity, which negatively interferes with daily life activities, causing poor academic performance, social difficulties, and greater health problems [15]. In general, oculomotor problems in the capacity and precision of fixations have been detected in all three NDD [16,17]. Children diagnosed with dyslexia often experience notable instability in their eye fixations and an increased frequency of regressions while reading. This population also tends to exhibit prolonged saccadic reaction times, which can adversely affect their reading speed and comprehension [18]. The difficulty in maintaining stable fixations can necessitate the re-reading of text, thereby slowing the reading process and impacting academic performance. In children with attention deficit hyperactivity disorder (ADHD), there are observed deficits in the ability to inhibit oculomotor responses. This lack of control over their eye movements results in erratic and involuntary saccades, which can disrupt their focus on visual targets and sustained attention during tasks [19,20]. Such oculomotor deficiencies contribute to frequent gaze shifts, complicating their engagement in activities requiring prolonged visual concentration. Similarly, children with developmental coordination disorder (DCD) face challenges in maintaining steady fixation, performing tracking tasks, and executing rapid saccades. These children tend to make more errors in anti-saccade tasks, where they must look away from a visual stimulus instead of directly at it. The increased error rate and difficulty in maintaining smooth, coordinated eye movements can hinder their ability to process visual information effectively, thereby affecting both academic performance and daily activities [21]. The combination of these oculomotor challenges poses significant barriers to participation in tasks that demand precise and coordinated eye movements. Antisaccades are voluntary eye movements where an individual must suppress a reflexive saccade towards a visual stimulus and instead look in the opposite direction. This task requires significant cognitive control, involving the inhibition of the automatic response and the generation of a deliberate saccade in the opposite direction. The objective of this study was to characterize the fixations during reading in children with NDD.

## 2. Materials and Methods

### 2.1. Patients

This was a prospective non-randomized comparative study evaluating a total of 68 children with ages ranging from 6 to 12 years old at the Department of Optometry of the Hospital Quirón Salud (Huesca, Spain). The research adhered to the principles of the Declaration of Helsinki and was approved by the ethics committee of the University of Valencia, Spain (2703478).

Informed written consent was obtained from the children’s parents after explaining the study protocol, risks, and benefits to them. Patient recruitment was carried out by two optometrists from the Optometry Unit at the Hospital Quirón Salud. Most of the children were referred by an ophthalmologist and had undergone a comprehensive ophthalmological examination. From the recruited patients, two groups were formed based on the following criteria: the inclusion criteria for all groups were children who were either corrected ametropic or emmetropic, achieving a corrected distance visual acuity (CDVA) of 0.00 logMAR (20/20 Snellen) or better. The exclusion criteria included any active ocular or systemic disease at the time of examination, any previous ocular surgery, prior vision therapy, and conditions such as amblyopia, strabismus, and nystagmus.

The sample comprised 24 students diagnosed with neurodevelopmental disorders (NDD) at their respective schools, with these diagnoses subsequently confirmed by a pediatric neurologist. The students were diagnosed based on the criteria specified in the Diagnostic and Statistical Manual of Mental Disorders, Fifth Edition (DSM-5). The specific disorders included dyslexia, attention deficit hyperactivity disorder (ADHD), and developmental coordination disorder (DCD). The diagnostic process involved standardized assessments conducted by school psychologists, followed by verification through clinical evaluations by a pediatric neurologist. Notably, none of the students were on medication for their conditions during the study. This thorough approach ensured the accurate identification of the type and severity of each disorder, allowing for a robust analysis of their association with oculomotor deficiencies. Among both groups—children with and without NDD—oculomotor dysfunction was diagnosed according to the guidelines defined by the Developmental Eye Movement (DEM) test. Thus, three groups were differentiated: a control group (CG) consisting of 20 children (*n* = 5 girls) aged 6 to 12 years, a group of 24 children (*n* = 4 girls) with NDD (NDDG), and a group of 24 children (*n* = 7 girls) with oculomotor anomalies but without NDD (OAG).

To ensure the robustness of our findings, we performed cross-validation by correlating the saccadic data obtained from the subjective NSUCO test with both the digitized NSUCO data and the eye tracker data. Specifically, we correlated the variables of saccadic ability and precision. Patients were shown points of 0.5 cm diameter appearing on the screen every 1 s for 5 cycles.

To determine the relationship between subjective and objective oculomotor evaluations, we used Spearman’s rank correlation coefficient, suitable due to the ordinal nature of the NSUCO data and the potential non-linearity. The ability evaluated by the NSUCO corresponds to the complete saccades recorded by the eye tracker, which refers to the capability to perform a full eye movement towards the fixation stimulus. This is equivalent to the ability variable in the NSUCO. Precision, as evaluated by the NSUCO, is equated with the number of hypometric saccades recorded by the eye tracker, meaning that the eye movements fall short of the target stimulus. We considered ρ values of 0.00 < ρ < 0.3 as weak, 0.3 < ρ < 0.6 as moderate, and ρ > 0.6 as strong. 

### 2.2. Examination Protocol

A complete visual examination was performed before the oculomotor analysis, which included the measurement of manifest and cycloplegic refraction, uncorrected distance visual acuity (UDVA) and corrected distance visual acuity (CDVA), and distance and near heterophoria (40 cm) using the cover test (with negative values representing exophoria and positive values representing esophoria); the measurement of the near point of convergence (NPC); and the measurement of distance and near stereopsis.

The evaluation of eye movements was carried out with three types of tests, two subjective—the Developmental Eye Movement (DEM) test and the Northeastern State University College of Optometry (NSUCO) test—and one objective, through the eye tracking system.

The “Developmental Eye Movement” (DEM) test is a standardized, subjective oculomotor assessment designed to easily identify small-amplitude saccades during reading. It is validated for children aged 6 to 14 years. The test consists of two sheets (A and B), each containing a vertical list of 40 numbers divided into two parts of 2 rows, followed by a horizontal list of another 80 numbers in 16 rows. The task involves reading the numbers on each sheet as quickly and accurately as possible. The time taken for each sheet, along with the number and types of errors, were recorded. Calculations were then performed to obtain a ratio (horizontal time/vertical time) and compare these values with age-specific norms.

The “Northeastern State University College of Optometry” (NSUCO) test is a subjective assessment that evaluates larger-amplitude saccadic movements. It examines three performance areas: ability, which assesses whether the patient can perform the task; accuracy, which evaluates how precisely the task is performed; and head movements, which assesses the head movements performed to complete the task. The test was conducted with the patient standing 40 cm in front of the examiner, using both eyes. Small colored spheres, 0.5 cm in diameter, mounted on a rod, were used as fixation stimuli as per the NSUCO guidelines. Saccadic movements were evaluated by asking the patient to alternate their fixation between two stimuli placed 20 cm apart horizontally. Based on the examiner’s observations, the scoring was determined according to the criteria outlined in Table 1 [17].

Objectively, the oculomotor assessment was performed with the Tobii Eye X Eye Tracker (Tobii, Stockholm, Sweden) and the Clinical Eye Tracking software by Thomson Software Solutions (Welham Green, UK). For such purposes, the patient sat 50 cm away from a 24-inch monitor. A study stimulus was chosen for the evaluation of short reading saccades, consisting of a specific text according to the patient’s age, composed of 81 words. The patient was instructed to read the text, and the device recognized the visual axes, showing the horizontal and vertical position of the eyes and vergences in real time, as shown in Figure 1 and Figure 2. The variables measured included the number of fixations per minute, average fixation duration, percentage of regressions, resolution time, words per minute, and mean number of fixations per line. The resolution time refers to the total duration required for a subject to read a given text from the beginning to the end, encompassing the entire period from the initial fixation on the first word to the final fixation on the last word. All variables were automatically identified and recorded by the Clinical Eye Tracking software.

### 2.3. Statistical Analysis

All data were compiled into an Excel Version 16.86 database. Various variables were collected for each subject during the visual assessment, and descriptive statistics were performed using these data. Statistical analysis was conducted using the IBM SPSS software (version 23.0; IBM Corporation, Somers, NY, USA) after importing the Excel database.

The normality of the variables was assessed using the Kolmogorov–Smirnov test. While some variables exhibited a normal distribution, others did not. Due to the sample size being less than 30, non-parametric tests were chosen for the statistical analysis, with the significance level set at a *p*-value < 0.05.

The Kruskal–Wallis test was employed to compare the quantitative variables studied. The post hoc analysis used to identify specific group differences utilized the Mann–Whitney test adjusted with Bonferroni correction. For qualitative variables, the chi-square test assessed distribution differences among groups, with *p*-values < 0.05 considered statistically significant.

The choice of post hoc test considered the influence of the sample size on the significance level outcomes, as highlighted by Juarros-Basterretxea et al. [22]. Factors such as homoscedasticity and balanced sample sizes across groups were crucial in selecting appropriate post hoc tests. Determining the magnitude of statistically significant differences included defining the sample size and assessing the effect size power to detect significant group disparities.

The relationship between the effect size and power implies that smaller sample sizes may yield higher η values and lower power, potentially indicating significant differences despite higher *p*-values. Effect sizes close to 0 indicate small effects, while those nearing 1 signify large effects. An effect size of 0.14 is moderate, and 0.34 is large. Power values between 0.50 and 0.80 are moderate, while values exceeding 0.80 are considered large.

## 3. Results

### 3.1. General Clinical Data

A total of 68 children (range, 6 to 12 years old), with a mean age of 8.64 (SD = 1.84), were enrolled, with three differentiated groups: CG (*n* = 20), OAG (*n* = 24), and NDDG (*n* = 24).

To conduct the study using the objective test, cross-validation was performed. For this purpose, we analyzed the correlation between the NSUCO test results and the eye tracker data. A significant, moderate-to-high correlation was observed between the NSUCO test’s assessment of saccadic ability, and the number of complete saccades recorded by the eye tracker (r = 0.702; *p* < 0.001). Additionally, a significant correlation was found between the NSUCO-evaluated precision and the number of hypometric saccades detected by the eye tracker (r = 0.686; *p* < 0.001). These findings are presented in Table 2, demonstrating that the subjective evaluations from the NSUCO test align well with the objective measurements from the eye tracker, supporting the validity of both assessment methods.

### 3.2. Oculomotor Analysis

Table 3 shows the visual characteristics in the control group, the group with oculomotor dysfunction, and the group with neurodevelopmental disorders included in the study. The variable “DEM type” is included in this table because this test was used for classification into the OAG sample groups, and, therefore, it could not be used as a variable in our comparisons, as this would introduce bias into the results. Statistically significant differences were found in the average near stereopsis, in the cover test at distance and near, and in the DEM type variable.

Table 4 and Table 5 show the analysis of the results obtained with the subjective oculomotor tests DEM and NSUCO in the three groups studied. Statistically significant differences were found in all characteristics evaluated with the two oculomotor tests. 

Table 6 summarizes the results of the objective eye movement analysis using the eye tracker across the three studied groups. Statistically significant differences were found in the resolution time, number of words per minute, and percentage of regressions. However, for the remaining variables, despite not showing significant differences (*p* > 0.05), clinical observations indicated a trend towards larger mean differences among the three groups (see Figure 1 and Figure 2). To statistically verify this difference, effect size and effect power tests were employed, as depicted in Table 7.

Figure 3 presents a boxplot illustrating the number of fixations per minute (fixations/min) for the different groups studied. It is important to note that the Y-axis represents the count of fixations per minute rather than a unit of time. This adjustment is made to correct the labeling discrepancy. The boxplot provides a summary of the distribution of the fixation counts using five key statistics: the minimum, the first quartile (Q1), the median (Q2), the third quartile (Q3), and the maximum. The central box indicates the interquartile range (IQR), which encompasses the middle 50% of the data, while the whiskers extend to the smallest and largest values within 1.5 times the IQR from the quartiles. The “X” mark within the box represents the mean fixation count. These elements collectively offer a visual representation of the central tendency and variability in the fixation counts per minute among the groups. This detailed explanation ensures that readers can accurately interpret the data presented in Figure 3.

Figure 4 shows a boxplot illustrating the mean duration of fixations for the different groups analyzed. The Y-axis represents the mean duration of fixations. This plot summarizes the distribution of fixation durations using five key statistics: the minimum value, the first quartile (Q1), the median (Q2), the third quartile (Q3), and the maximum value. The central box represents the interquartile range (IQR), which contains the middle 50% of the data, while the whiskers extend to the lowest and highest values within 1.5 times the IQR from the quartiles. The “X” mark inside the box indicates the mean fixation duration.

In the case of the number of words per minute, the effect size was moderate, but the power was high, so there was a high probability of detecting differences between the groups despite the *p*-value > 0.05 (*p*-value = 0.011).

For the fixations per minute, the *p*-value was >0.05 (*p*-value = 0.223) but the effect size was very large (η = 0.33) and the power was low (power = 0.239). This may be due to the small sample size, which prevented us from finding significant differences.

Regarding the percentage of regressions, there was a statistically significant difference (*p*-value < 0.001), a very large effect size (η = 0.334), and very high power (power = 0.999). Therefore, there was a very high probability of detecting differences compared to the healthy controls in the regressions.

## 4. Discussion

The findings of this study provide evidence of a significant correlation between the subjective assessments of the NSUCO test and the objective measurements obtained via an eye tracker. Specifically, the correlation between the NSUCO-assessed saccadic ability and the number of complete saccades recorded by the eye tracker indicates that the NSUCO is a reliable predictor of complete saccades, a key indicator of oculomotor function. This suggests that as the NSUCO-assessed ability increases, so does the number of complete saccades detected.

Furthermore, the significant correlation between the NSUCO-assessed precision and the number of hypometric saccades detected by the eye tracker highlights the NSUCO’s capacity to reflect the saccadic precision, despite being a subjective measure. These results align with previous research that found relationships between subjective and objective oculomotor function measures [23], supporting the concurrent validity of both assessment methods. However, the limitations include the non-representative sample, which may affect the generalizability of the results, and the fact that a correlation does not imply causation, necessitating further studies to explore these relationships.

The study of fixations or saccadic movements provides important clues in the diagnosis of oculomotor dysfunction. However, the established diagnostic criteria remain unvalidated on a large scale. In this study, the DEM test was used to detect subjects with oculomotor anomalies, as it shows good consistency in classifying patients as either passing or failing the test [24]. Therefore, oculomotor evaluation was combined with the NSUCO test, which assesses the control of long saccades [25]. Although this test has limitations, such as those related to examiner control, language, or patient attention, research has focused on using eye tracking to eliminate the biases caused by examiner subjectivity [26,27].

In our series, oculomotor function was studied in a group of healthy children, another group with oculomotor anomalies, and another group with specific learning disorders such as dyslexia, ADHD, or DCD. Several studies have previously reported that children with learning difficulties can have impaired oculomotor function, which negatively influences the reading process [21,28,29,30]. Our aim was to investigate specifically the oculomotor pattern during the reading process in order to obtain a better understanding of the potential impact of oculomotor dysfunction in children with NDD on their reading performance.

Regarding the evaluation of fixations with the NSUCO test, there were significantly lower scores in OAG and NDDG compared to CG. Similarly, for ability, precision, and head movements during smooth pursuit and saccades, significantly lower scores were found in OAG and NDDG compared to GC, but there were no significant differences when comparing results of OAG and NDDG. This suggests that oculomotor dysfunction may be a common feature in children with various neurological conditions, as well as in children without specific diagnoses of neurodevelopmental disorders. Children with OAG tended to score lower on the NSUCO compared to the control group (CG). These results are consistent with those of Maples et al. [26], who explained the development and standardization of the NSUCO test and established a correlation between the age and test results. Bucci et al. [31] reported that there were statistically significant differences between a CG, an OAG, and a NDDG in the NSUCO test, obtaining lower scores in cases of learning difficulties.

Regarding the DEM test, there were no significant differences observed between the control group (CG) and the oculomotor anomaly group (OAG) in their vertical reading times (Cards A and B). However, significant differences in this parameter were found between the control group (CG) and the neurodevelopmental disorder group (NDDG). Specifically, children with NDD tended to exhibit longer vertical reading times compared to those in the OAG, with the shortest times observed in the CG. Concerning Card C and the ratio, the results obtained in the CG significantly differed from those obtained in both the OAG and NDDG. Additionally, there were statistically significant differences in the number of errors among the three groups involved in the study. This is consistent since subjects with neurodevelopmental disorders tended to exhibit a greater number of errors in the test and had higher horizontal reading times than healthy children. These results are consistent with those obtained by Tiadi et al. [28] and Raghuram et al. [29], which indicated that children with dyslexia took longer to read Card C compared to children without specific learning disorders. Similar findings were also reported by our research group in a previous study [2]. While it is well established that children with dyslexia face challenges in reading tasks, our study provides novel insights into the specific oculomotor dysfunction contributing to these difficulties. By using both subjective and objective measures, we were able to identify distinct patterns of poor fixation stability, increased regressions, and abnormal saccadic movements that underline reading impairments in dyslexic children. These findings not only corroborate previous research but also enhance our understanding of the underlying mechanisms affecting reading performance in children with dyslexia.

Apart from oculomotor issues, previous studies have demonstrated that children with neurodevelopmental disorders also commonly experience binocular problems [20,30]. In our study, statistically significant differences were observed among the control group (CG), neurodevelopmental disorder group (NDDG), and oculomotor anomaly group (OAG) regarding stereopsis, with children diagnosed with neurodevelopmental disorders showing poorer stereopsis values. According to the research conducted by Buzelli et al. [30], stereopsis and visual acuity (VA) were comparable between children with and without dyslexia. Within our cohort, significant differences in the magnitude of phoria at the binocular level were noted between the CG and OAG, with subjects in the OAG tending towards exophoria. Essentially, children with oculomotor anomalies are more likely to exhibit exophoria. These findings are consistent with those reported by Muzaliha et al. [32], who found that 30% of children with dyslexia exhibited convergence insufficiency and poor accommodative abilities.

Deficiencies in the inhibition of the oculomotor response have been observed in children with ADHD [25]. Additionally, children with DCD exhibit difficulties in maintaining fixation and in tracking tasks and high-speed saccades, along with more errors in antisaccades. Recent studies have shown that children with DCD exhibit significant impairments in feedforward action control during visually guided upper limb movements, such as pointing and reaching actions, and demonstrate delays in attentional disengagement and motor initiation compared to typically developing controls [33]. These findings highlight the need to consider the specific neurophysiological and motor control challenges faced by children with different types of neurodevelopmental disorders when assessing and addressing oculomotor deficiencies.

John Stein’s magnocellular theory suggests that dyslexia is linked to dysfunction in the magnocellular pathway, which is essential in processing motion and controlling eye movements. Stein argues that individuals with dyslexia have impairments in their visual magnocellular system, leading to challenges with visual motion sensitivity and binocular stability, both critical for effective reading. These deficits can cause the visual perceptual instability frequently observed in dyslexic individuals, where letters may appear to shift or blur. This instability can lead to common reading issues such as letter reversals and misordering. Stein further notes that genetic and environmental factors might affect the development of magnocellular neurons, connecting the neurobiological aspects to the reading difficulties encountered by dyslexic readers [34].

Regarding the oculomotor evaluation using objective eye tracker systems, statistically significant differences were found between the CG and NDDG in the test resolution time and words per minute, indicating that subjects with neurodevelopmental disorders tended to take longer to read the text compared to healthy subjects. There were also significant differences in the regressions among the three groups, suggesting a very high probability of detecting differences compared to healthy subjects.

For the mean fixation duration and fixations per minute, no significant differences were found between the groups; however, the effect size was large and the power was low. This could be due to the small sample size, but it is important to highlight because there was a clinical difference observed between the groups, with a trend for subjects with neurodevelopmental disorders to have longer fixation durations. This justifies the need for further research in this area and an increase in the sample size, as these parameters are likely to show significant differences in future studies with larger samples.

Other authors, such as Raghuram et al. [28], Bucci et al. [31], Cadani S et al. [35], Sessau et al. [36] Vagge A et al. [37], Molina et al. [38], Sherigar et al. [39], Bonifacci et al. [40], and Lee et al. [41], found that children with neurodevelopmental disorders took longer to read the text and exhibited a greater number of fixations and regressions than normal children. In a study conducted by Lee et al. [40], the same results were obtained using an eye tracker, and, additionally, after undergoing visual therapy training, the children achieved more stable fixations and improved fixation durations. On the other hand, according to Raghuram et al. [29], children with dyslexia have oculomotor problems, but it is unclear whether this is a cause or effect of the reading impairment. Sumner et al. [42] studied eye movements in children with DCD and observed that the fixations were very imprecise, suggesting that they might be caused by the presence of intrusive saccades. Furthermore, Razuk et al. [43] proposed that the presence of longer fixation durations in children with neurodevelopmental disorders may be due to the need for a longer pause to perform semantic processing or a difficulty in decoding words or syllables. Other authors, such as Bucci et al. [31], suggest that deficits in oculomotor behavior in children with dyslexia may be due to the immaturity of the mechanism responsible for the interaction between the saccadic and vergence systems.

This study acknowledges several limitations. Firstly, the sample size was relatively small; however, the aim was to identify trends that could be explored further in future studies with larger samples. Nonetheless, significant differences were observed between the control group (CG) and neurodevelopmental disorder group (NDDG) in some of the assessed oculomotor parameters. Secondly, the age range of the participants should be considered, as younger children are more prone to attention lapses, particularly during tasks like reading. Additionally, the developmental stages of reading vary, with six year olds beginning with basic word or phrase comprehension, while twelve year olds develop analytical skills to understand more complex texts [44]. Furthermore, the specific types of neurodevelopmental disorders within the groups were not differentiated. Future research should consider studying fixations across different age groups. Finally, although the eye tracking methodology is suitable for the assessment of eye movements due to its objectivity, standardized data that differentiate between various types of neurodevelopmental disorders are still needed to improve the diagnostic and therapeutic processes.

## 5. Conclusions

Children with neurodevelopmental disorders exhibit oculomotor impairments, although this sign is not specific to this population, as these impairments can also be present in children without specific learning disorders. Regressions, longer fixation durations, and longer reading times appear to be characteristic oculomotor signs in children with TNND. Additionally, they also take longer to complete the DEM test, with a higher number of errors and higher ratio values.

For their detection, objective eye tracking systems are of great interest because they provide a suitable tool for the characterization of oculomotor anomalies due to their objective nature in exploration. However, subjective systems like the NSUCO or DEM are also effective in assessing eye movements and are easily applicable in clinical optometric settings for children with specific learning disorders.

## Figures and Tables

**Figure 1 brainsci-14-00750-f001:**
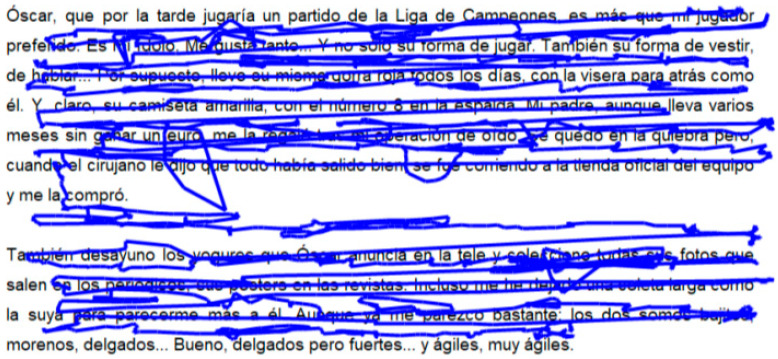
Reading text chosen to evaluate short saccades. The blue line represents the movement of the subject’s fixation throughout the reading of the text.

**Figure 2 brainsci-14-00750-f002:**
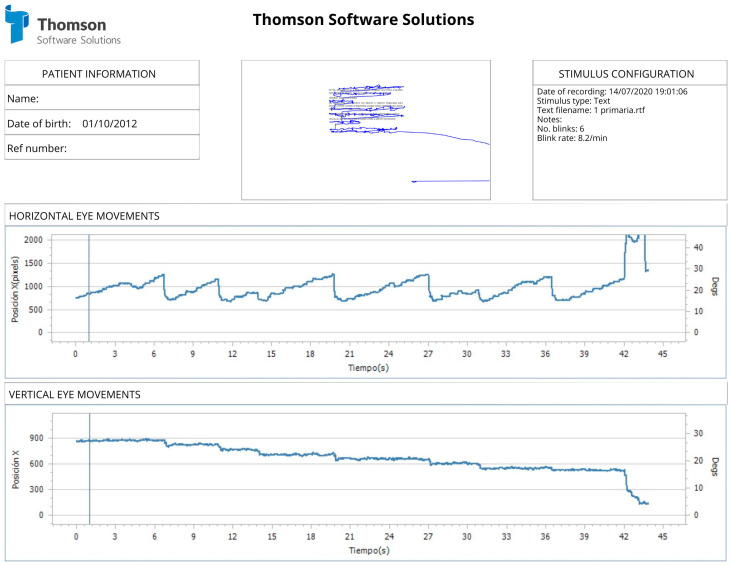
Representative figure of eye tracking during reading with the Eye tracker. The first graph shows the horizontal tracking of the visual axes. The second graph represents the vertical visual tracking of the visual axes.

**Figure 3 brainsci-14-00750-f003:**
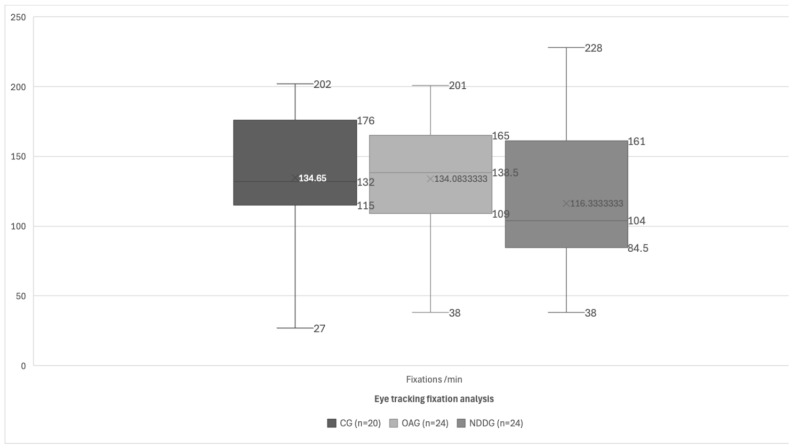
Diagram showing the statistically significant differences found between the control group (CG), oculomotor anomaly group (OAG), and neurodevelopment disorder group (NDDG) in the fixations/min detected with the eye tracker.

**Figure 4 brainsci-14-00750-f004:**
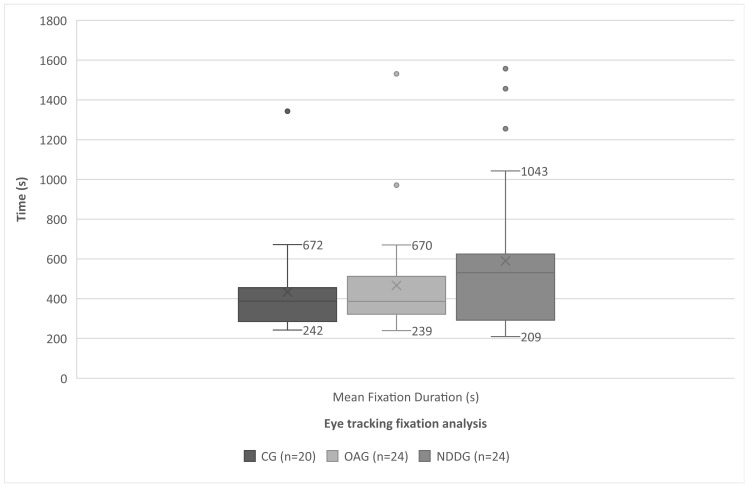
Diagram showing the statistically significant differences found between the control group (CG), oculomotor anomaly group (OAG), and neurodevelopmental disorder group (NDDG) in the mean fixation duration (s) detected with the eye tracker.

**Table 1 brainsci-14-00750-t001:** Scoring procedure used in the NSUCO test for the evaluation of oculomotor function.

Performance Area	Evaluation Procedure	Scoring System
Ability	Patient’s ability to perform 5 cycles of change in fixation between two stimuli presented	1 point: 1 cycle or no ability 2 points: 2 cycles 3 points: 3 cycles 4 points: 4 cycles 5 points: 5 cycles
Accuracy	Patient’s ability to perform 5 cycles of change in fixation without performing refixations	1 point: significant hyper- or hypometric movements 2 points: large to moderate hyper- or hypometric movements 3 points: slight hyper- or hypometric movements but constant 4 points: slight hyper- or hypometric movements but intermittent 5 points: no correcting refixations
Head movement associated	Patient’s ability to perform 5 cycles of change in fixation without head or body movements	1 point: 1 cycle or no ability 2 points: 2 cycles 3 points: 3 cycles 4 points: 4 cycles 5 points: 5 cycles

**Table 2 brainsci-14-00750-t002:** Spearman’s correlation coefficients and *p*-values for the relationship between NSUCO ability, NSUCO accuracy, and eye tracking measures.

Correlation, *n* = 68	ρ Spearman	*p*-Value
NSUCO ability + 5 rounds completed (objetive measure)	0.702	0.00 (0.00)
NSUCO accuracy + hypometric saccades (eye tracking)	0.686	0.00 (0.00)

**Table 3 brainsci-14-00750-t003:** Summary of the main visual characteristics of the three groups evaluated in the study: CG, control group; OAG, group of children with oculomotor abnormalities; NDDG, group of children with neurodevelopmental disorders.

Mean (SD) Median (Range)	CG (*n* = 20)	OAG (*n* = 24)	NDDG (*n* = 24)	*p*-Value
Age (years)	8.35 (2.25) 8.00 (6 to 11)	8.87 (1.84) 9.00 (6 to 12)	8.70 (1.45) 9.00 (6 to 11)	0.499
Sphere RE (D)	0.00 (0.00) 0.00 (−0.75 to +3.00)	0.00 (0.00) 0.00 (0.00 to 0.00)	0.14 (1.18) 0.00 (0.00 to +0.50)	0.837
Cylinder RE (D)	−0.21 (0.773) 0.00 (−3.25 to 0.00)	0.00 (0.00) 0.00 (0.00 to 0.00)	−0.14 (0.43) 0.00 (−1.50 to 0.00)	0.180
Sphere LE (D)	0.08 (0.72) 0.00 (0.00 to +3.50)	0 (0.00) 0.00 (0.00 to 0.00)	0.16 (1.21) 0.00 (0.00 to +5.00)	0.748
Cylinder LE (D)	−0.16 (0.71) 0.00 (−3.50 to 0.00)	0.00 (0.00) 0.00 (0.00 to 0.00)	−0.11 (0.37) 0.00 (−1.50 to 0.00)	0.301
LogMAR CDVA (RE)	0.00 (0.00) 0.00 (0.00 to 0.00)	0.00 (0.00) 0.00 (0.00 to 0.00)	0.00 (0.00) 0.00 (0.00 to 0.00)	0.156
LogMAR CDVA (LE)	0.00 (0.00) 0.00 (0.00 to 0.00)	0.00 (0.00) 0.00 (0.00 to 0.00)	0.00 (0.00) 0.00 (0.00 to 0.00)	0.156
Near Stereopsis (sec arc)	31.00 (13.73) 20.00 (20.00 to 60.00)	38.08 (18.81) 40.00 (20 to 100)	72.46 (79.46) 45.00 (20.00 to 400.00)	0.002 * CG-OAG 0.193 CG-NDDG 0.001 * OAG-NDDG 0.014 *
NPC Break (cm)	4.55 (5.70) 3.00 (0.00 to 20.00)	6.83 (5.94) 7.00 (0.00 to 20.00)	7.71 (5.86) 8.00 (0.00 to 20.00)	0.121
NPC Recovery (cm)	4.20 (5.37) 2.00 (0.00 to 18.00)	6.54 (5.89) 6.00 (0.00 to 18.00)	7.46 (6.63) 7.00 (0.00 to 25.00)	0.173
Near Cover Test (∆)	−1.05 (2.23) 0.00 (−6.00 to 0.00)	−3.87 (4.20) −4.00 (−15.00 to 0.00)	−3.17 (5.09) −2.00 (−14.00 to 0.00)	0.043 * CG-OAG 0.012 * CG-NDDG 0.052 OAG-NDDG 0.677
DEM_Type	1.0 (0.00) 1.0 (1.00 to 1.00)	2.45 (0.83) 2.00 (2.00 to 4.00)	2.91 (1.08) 3.00 (1.00 to 4.00)	<0.001 * CG-OAG < 0.001 * CG-NDDG < 0.001 * OAG-NDDG 0.032 *

SD: standard deviation, RE: right eye, LE: left eye, D: diopter, CDVA: corrected distance visual acuity, NPC: near point of convergence (cm). The results of the cover test are expressed as negative in the presence of exophoria and positive in the presence of esophoria. * *p*-values representing statistically significant differences (*p*-value < 0.05).

**Table 4 brainsci-14-00750-t004:** Summary of the DEM test results in the three groups evaluated in the study: CG, control group; OAG, group of children with oculomotor abnormalities; NDDG, group of children with neurodevelopmental disorders.

Mean (SD) Median (Range)	CG (*n* = 20)	OAG (*n* = 24)	NDDG (*n* = 24)	*p*-Value
Time Sheet A (s)	22.10 (4.87) 20.00 (15.00 to 34.00)	23.29 (5.60) 23.00 (15:00 to 40.00)	28.58 (7.78) 29.00 (15:00 to 42.00)	0.005 * CG-OAG 0.362 CG-NDDG 0.002 * OAG-NDDG 0.018 *
Time Sheet B (s)	23.65 (4.98) 22.50 (15.00 to 42.00)	25.96 (5.83) 25.00 (18.00 to 39.00)	32.16 (9.80) 30.50 (19.00 to 55.00)	0.004 * CG-OAG 0.226 CG-NDDG 0.002 * OAG-NDDG 0.022 *
Time Sheet C (s)	64.10 (19.70) 59.00 (43.00 to 110.00)	91.08 (29.25) 78.00 (49.00 to 166.00)	114.75 (50.66) 105.00 (37.00 to 212.00)	<0.001 * CG-OAG 0.001 * CG-NDDG < 0.001 * OAG-NDDG 0.097
DEM Ratio	1.39 (0.21) 1.35 (1.02 to 1.70)	1.85 (0.45) 1.80 (1.03 to 2.92)	1.94 (0.92) 1.70 (0.91 to 5.12)	0.001 * CG-OAG < 0.001 * CG-NDDG 0.007 * OAG-NDDG 0.509
Number of Errors	0.70 (1.081) 0.00 (0.00 to 3.00)	5.00 (4.41) 4.00 (0.00 to 20.00)	10.09 (10.62) 6.00 (1.00 to 45.00)	<0.001 * CG-OAG < 0.001 * CG-NDDG < 0.001 * OAG-NDDG 0.044 *

SD: standard deviation, s: second. * *p*-values representing statistically significant differences (*p*-value < 0.05).

**Table 5 brainsci-14-00750-t005:** Summary of the NSUCO test results in the three groups evaluated in the study: CG, control group; OAG, group of children with oculomotor abnormalities; NDDG, group of children with neurodevelopmental disorders.

Mean (SD) Median (Range)	CG (*n* = 20)	OAG (*n* = 24)	NDDG (*n* = 24)	*p*-Value
Fixation	4.85 (0.36) 5.00 (3.00 to 5.00)	3.12 (0.79) 3.00 (1.00 to 5.00)	3.58 (1.38) 4.00 (1.00 to 4.00)	<0.001 * CG-OAG < 0.001 * CG-NDDG 0.010 * OAG-NDDG 0.014 *
NSUCO test: saccades				
Ability	4.80 (0.41) 5.00 (3.00 to 5.00)	2.33 (0.70) 2.00 (1.00 to 4.00)	2.37 (0.767) 2.00 (1.00 to 4.00)	<0.001 * CG-OAG < 0.001 * CG-NDDG < 0.001 * OAG-NDDG 0.940
Precision	4.75 (0.44) 5.00 (3.00 to 5.00)	2.33 (0.82) 2.00 (1.00 to 4.00)	2.33 (0.82) 2.00 (1.00 to 4.00)	<0.001 * CG-OAG < 0.001 * CG-NDDG < 0.001 * OAG-NDDG 0.689
Head/Body Movement	4.80 (0.410) 5.00 (3.00 to 5.00)	2.25 (0.79) 2.00 (1.00 to 4.00)	2.00 (0.83) 2.00 (1.00 to 4.00)	<0.001 * CG-OAG < 0.001 * CG-NDDG < 0.001 * OAG-NDDG 0.680

SD: standard deviation. * *p*-values representing statistically significant differences (*p*-value < 0.05)

**Table 6 brainsci-14-00750-t006:** Summary of the eye tracker results in the three groups evaluated in the study: CG, control group; OAG, group of children with oculomotor abnormalities; NDDG, group of children with neurodevelopmental disorders.

Mean (SD) Median (Range)	CG (*n* = 20)	OAG (*n* = 24)	NDDG (*n* = 24)	*p*-Value
Resolution Time (s)	67.75 (25.31) 57.50 (34.80 to 142.60)	80.86 (44.04) 74.35 (31.90 to 248.10)	**103.71 (51.49)** 153.00 (36.80 to 286.00)	0.011 * CG-OAG 0.225 **CG-NDDG 0.003 *** OAG-NDDG 0.068
Words Per Minute	79.85 (25.03) 84.52 (34.08 to 139.66)	73.85 (33.34) 65.30 (19.59 to 152.35)	**57.02 (25.26)** 56.23 (19.38 to 132.07)	0.011 * CG-OAG 0.225 **CG-NDDG 0.003 *** OAG-NDDG 0.068
No. Fixations	144.85 (69.65) 125.50 (32.00 to 286)	167.25 (72.87) 150.00 (41.00 to 323.00)	167.00 (56.88) 153.00 (80.00 to 279.00)	0.329
Fixations/min	134.65 (47.88) 132.00 (27.00 to 202.00)	134.08 (42.32) 138.50 (60.00 to 201.00)	**116.33 (51.65)** 104.00 (38.00 to 228.00)	0.223
Mean Fixation Duration (s)	433.75 (242.96) 387.50 (242.00 to 1343.00)	**466.66 (278.59)** 386.00 (246.00 to 1531.00)	**589.66 (376.747)** 530.50 (209.00 to 1557.00)	0.224
Mean No. Fixations/Row	16.20 (35.09) 15.96 (9.50 to 26.75)	19.69 (6.25) 18.63 (11.44 to 31.29)	17.86 (6.58) 16.00 (11.17 to 42.50)	0.137
Percentage of Regressions (%)	24.03 (3.78) 24.10 (15.30 to 31.70)	31.72 (36.82) 31.80 (19.40 to 48.80)	33.13 (35.47) 31.80 (20.90 to 45.00)	<0.001 * **CG-OAG < 0.001 *** **CG-NDDG < 0.001 *** OAG-NDDG 0.370

SD: standard deviation, No.: number, s: second, %: percentage, min: minutes. * *p*-values representing statistically significant differences (*p*-value < 0.05). Non-significant values with large mean differences are highlighted in bold.

**Table 7 brainsci-14-00750-t007:** Summary of the effect size and power of the effect results.

	η^2^	Power
Resolution Time (s)	0.111 *	0.705 *
Words Per Minute	0.108 *	0.690 *
No. Fixations	0.024	0.179
Fixation/min	0.33 *	0.239 *
Mean Fixation Duration (s)	0.047	0.330
Mean No. Fixations/Row	0.053	0.366
Percentage of Regressions (%)	0.334 *	0.999 *

SD: standard deviation, No.: number, s: second, %: percentage, min: minutes. * indicates large effect sizes.

## Data Availability

Data available on request due to privacy and ethical restrictions.

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
