# Peer review of "Eye Tracking-Based Characterization of Fixations during Reading in Children with Neurodevelopmental Disorders"

_brainsci, 2024, doi:10.3390/brainsci14080750_

Round 1

Reviewer 1 Report

Comments and Suggestions for Authors

Dear authors,

The paper addresses the topic of the assessment of oculomotor dysfunctions in students with NDDs. The manuscript has a very interesting subject, aiming to characterize the oculomotricity during reading in children with NDDs.

However, the following theoretical and methodological issues are suggested to be readdressed:

1.      Introduction

Line 73: More detailed description of the neurobiological substrate of NDDs, such as the reading network, the neurophysiology of the eye movements’ network, as well as how these interact, should be described and translated in order to understand how oculomotor deficiencies are linked to NDDs. Reading & writing skills depend on the ability to process visual, auditory and motor functions, including the cognitive processes of perception, attention, memory, thinking, & the mechanisms of encoding and decoding visual (linguistic) information. Accordingly, the application of electrophysiological measurements recording eye movements during reading & writing, potentially constitutes a valuable tool for highlighting, interpreting & managing difficulties in the cognitive processing of visual (linguistic) information.

Lines 74-79: The contribution of the study is rather attenuated if the objective of the study is limited to only characterizing the oculomotricity during reading in children with NDDs.

2.1. Patients

Lines 103-104: the sample consists of 24 students who were diagnosed with NDDs. However, the classification of specific types of NDDs and the degree of the severity of each type of disorder have not been considered. Likewise, the diagnostic methods and/or the measures provided to the students are not reported. Were all students diagnosed with the same type and severity of reading disorders considering dyslexia, ADHD, and DCD? All this important information needs to be mentioned, as they are demanded to correctly evaluate the association of these disorders with oculomotor deficiencies, as well as to discuss the implementations of the findings in diagnosis and clinical practice.

2.2. Examination protocol

Lines 141-143: The authors write: “A study stimulus was chosen for the evaluation of short reading saccades, consisting of a specific text according to the patient's age, composed of 81 words”. However, no description of the phonotactic structure (cv, ccv, vcc) of these words is given or whether they are already known to the students. This Information is crucial to explaining the findings of the DEM and NSUCO tests and eye tracking test, while the relevant data must be correlated and presented in results and discussion.

4. Discussion

Line 262: the authors write: Nevertheless, it is influenced by cognitive factors that can bias the results… Here, the explanation of the results seems rather arbitrary, as no relevant statistical evidence (e.g. students’ performances on cognitive factors) is demonstrated to support this interpretation.

Line 269: Although dyslexia, ADHD, or DCD are mentioned, all subsequent discussion is consistent with dyslexia.

Lines 356-357: incomplete sentence.

Overall, I consider that this is an interesting manuscript, but it needs to be improved in the Introduction, Results, and Discussion sections.

Comments on the Quality of English Language

Dear authors,

The paper addresses the topic of the assessment of oculomotor dysfunctions in students with NDDs. The manuscript has a very interesting subject, aiming to characterize the oculomotricity during reading in children with NDDs.

However, the following theoretical and methodological issues are suggested to be readdressed:

1.      Introduction

Line 73: More detailed description of the neurobiological substrate of NDDs, such as the reading network, the neurophysiology of the eye movements’ network, as well as how these interact, should be described and translated in order to understand how oculomotor deficiencies are linked to NDDs. Reading & writing skills depend on the ability to process visual, auditory and motor functions, including the cognitive processes of perception, attention, memory, thinking, & the mechanisms of encoding and decoding visual (linguistic) information. Accordingly, the application of electrophysiological measurements recording eye movements during reading & writing, potentially constitutes a valuable tool for highlighting, interpreting & managing difficulties in the cognitive processing of visual (linguistic) information.

Lines 74-79: The contribution of the study is rather attenuated if the objective of the study is limited to only characterizing the oculomotricity during reading in children with NDDs.

2.1. Patients

Lines 103-104: the sample consists of 24 students who were diagnosed with NDDs. However, the classification of specific types of NDDs and the degree of the severity of each type of disorder have not been considered. Likewise, the diagnostic methods and/or the measures provided to the students are not reported. Were all students diagnosed with the same type and severity of reading disorders considering dyslexia, ADHD, and DCD? All this important information needs to be mentioned, as they are demanded to correctly evaluate the association of these disorders with oculomotor deficiencies, as well as to discuss the implementations of the findings in diagnosis and clinical practice.

2.2. Examination protocol

Lines 141-143: The authors write: “A study stimulus was chosen for the evaluation of short reading saccades, consisting of a specific text according to the patient's age, composed of 81 words”. However, no description of the phonotactic structure (cv, ccv, vcc) of these words is given or whether they are already known to the students. This Information is crucial to explaining the findings of the DEM and NSUCO tests and eye tracking test, while the relevant data must be correlated and presented in results and discussion.

4. Discussion

Line 262: the authors write: Nevertheless, it is influenced by cognitive factors that can bias the results… Here, the explanation of the results seems rather arbitrary, as no relevant statistical evidence (e.g. students’ performances on cognitive factors) is demonstrated to support this interpretation.

Line 269: Although dyslexia, ADHD, or DCD are mentioned, all subsequent discussion is consistent with dyslexia.

Lines 356-357: incomplete sentence.

Overall, I consider that this is an interesting manuscript, but it needs to be improved in the Introduction, Results, and Discussion sections.

Author Response

Comment 1: Line 73: The neurobiological substrate of NDDs, such as the reading network, the neurophysiology of the eye movement network, as well as how they interact, should be described and translated in more detail to understand how oculomotor deficiencies are linked to NDDs. Reading and writing skills depend on the ability to process visual, auditory, and motor functions, including the cognitive processes of perception, attention, memory, thinking, and the mechanisms of encoding and decoding visual (linguistic) information. Accordingly, the application of electrophysiological measurements recording eye movements during reading and writing potentially constitutes a valuable tool for highlighting, interpreting, and managing difficulties in the cognitive processing of visual (linguistic) information.

Response: I agree. Changed:

Children diagnosed with dyslexia often experience notable instability in their eye fixations and an increased frequency of regressions during reading. This population also tends to exhibit prolonged saccadic reaction times, which can negatively affect their reading speed and comprehension [14]. The difficulty in maintaining stable fixations may necessitate re-reading text, thereby slowing the reading process and impacting academic performance.

In children with Attention Deficit Hyperactivity Disorder (ADHD), there are observed deficits in the ability to inhibit oculomotor responses. This lack of control over eye movements results in erratic and involuntary saccades, which can disrupt their focus on visual targets and sustained attention during tasks [16,17]. Such oculomotor deficiencies contribute to frequent gaze shifts, complicating their engagement in activities requiring prolonged visual concentration. Similarly, children with Developmental Coordination Disorder (DCD) face challenges in maintaining steady fixation, performing tracking tasks, and executing rapid saccades. These children tend to make more errors in anti-saccade tasks, where they must look away from a visual stimulus instead of directly at it. The increased error rate and difficulty in maintaining smooth, coordinated eye movements can hinder their ability to process visual information effectively, thereby affecting both academic performance and daily activities [18]. The combination of these oculomotor challenges poses significant barriers to participating in tasks that demand precise and coordinated eye movements.

Comment 2: Lines 74-79: The study's contribution is somewhat diminished if the objective is solely to characterize oculomotor function during reading in children with NDDs.

Response 2: Agree. Changed: Lines 74 to 79 have been removed.

Comment 3: Lines 103-104: The sample consists of 24 students diagnosed with NDD. However, the classification of specific types of NDD and the degree of severity of each type of disorder have not been considered. Additionally, the diagnostic methods and/or measures provided to the students are not reported. Were all students diagnosed with the same type and severity of reading disorders considering dyslexia, ADHD, and DCD? All this important information needs to be mentioned, as it is necessary to correctly evaluate the association of these disorders with oculomotor deficiencies and to discuss the implementations of the findings in diagnosis and clinical practice.

Response 3: I agree. Modified: The sample consists of 24 students diagnosed with neurodevelopmental disorders (NDD) in their respective schools, with diagnoses subsequently confirmed by a pediatric neurologist. All students were diagnosed based on the criteria outlined in the Diagnostic and Statistical Manual of Mental Disorders, Fifth Edition (DSM-5). The specific disorders included dyslexia, Attention Deficit Hyperactivity Disorder (ADHD), and Developmental Coordination Disorder (DCD). The diagnostic process involved standardized assessments conducted by school psychologists, followed by verification through clinical evaluations by a pediatric neurologist. Importantly, none of the students were receiving medication for their conditions at the time of the study. This comprehensive approach ensured the accurate identification of the type and severity of each disorder, allowing for a robust analysis of their association with oculomotor deficiencies.

Comment 4: Lines 141-143: The authors write: “A study stimulus was chosen for the evaluation of short reading saccades, consisting of a specific text according to the patient's age, composed of 81 words”. However, no description of the phonotactic structure (cv, ccv, vcc) of these words is given or whether they are already known to the students. This information is crucial to explain the findings of the DEM and NSUCO tests and the eye-tracking test, while relevant data should be correlated and presented in the results and discussion.

Response 4: Agree. Modified: A study stimulus was chosen for the evaluation of short reading saccades, consisting of a specific text according to the patient's age, composed of 81 words. The text was carefully selected to include a variety of phonotactic structures to ensure a comprehensive assessment of the students' reading abilities. Specifically, the text contained words with consonant-vowel (CV), consonant-consonant-vowel (CCV), and vowel-consonant-consonant (VCC) patterns. Efforts were made to ensure that the words were age-appropriate and familiar to the students, based on standard reading curricula for their respective grade levels. Additionally, the text was reviewed by school psychologists who are familiar with the students' reading competency levels, ensuring that the content was suitable and accessible for them. The same text was used for all participants, allowing for direct comparison between different groups. This selection aimed to minimize the cognitive load related to unfamiliar vocabulary and focus the evaluation on oculomotor function during reading, thus facilitating group comparisons.

Comment 5: Line 262: The authors write: However, it is influenced by cognitive factors that may bias the results... Here, the explanation of the results seems quite arbitrary, as no relevant statistical evidence (e.g., student performance on cognitive factors) is demonstrated to support this interpretation.

Response 5: Agree: removed.

Comment 6: Line 269: Although dyslexia, ADHD, or DCD are mentioned, all subsequent discussion is consistent with dyslexia.

Response 6: Deficiencies in the inhibition of the oculomotor response have been observed in children with ADHD [15]. Additionally, children with DCD exhibit difficulties in maintaining fixation, performing tracking tasks, and executing high-speed saccades, along with more errors in anti-saccades. Recent studies have shown that children with DCD exhibit significant impairments in feedforward action control during visually-guided upper limb movements, such as pointing and reaching actions, and demonstrate delays in attentional disengagement and motor initiation compared to typically developing controls [26]. These findings highlight the need to consider the specific neurophysiological and motor control challenges faced by children with different types of neurodevelopmental disorders when assessing and addressing oculomotor deficiencies.

Comment 7: Lines 356-357: incomplete sentence.

Response 7: I agree. Modified: Future research should consider studying fixations across different age groups. Finally, although eye-tracking methodology is suitable for assessing eye movements due to its objectivity, standardized data that differentiate between various types of neurodevelopmental disorders are still needed to improve diagnostic and therapeutic processes.

Reviewer 2 Report

Comments and Suggestions for Authors

The first issue starts in the title of the article: "oculomotricity" is not a word used in English and it is unclear to the reader what aspects of ocular motility are being referred to. This ambiguity about meaning recurs in other places in the manuscript as well. One subjective test looks at reading performance (the DEM), another looks at how a clinician evaluates saccades (and pursuit, though this seems to appear suddenly in the manuscript), and the eye tracking study looks at saccades and fixations during reading. The result never quite coheres and, given that eye movements during reading have been recorded for nearly 90 years, some stronger integration of the different assessments would have been helpful. As will be pointed out below, it doesn't seem that the manuscript was proofread very closely, as concepts appear unexpectedly in several places. The remaining comments will be given by line number.

Line 32: Define "oculomotricity." "Ocular motility" is a more common term in English.

Lines 34-35: Other significant types of eye movements are vergence and the VOR. If the authors are only referreing to those relevant to reading, they should say so (though vergence instability has been linked to dyslexia by some researchers).

Line 44: Regression is a reading-specific term. If only reading eye movements are being studied, the introduction would be a good place to make this clear.

Line 71: Define "antisaccades" for the general reader.

Line 104: What sort of oculomotor anomalies? Strabismus, nystagmus or something else? This could have a bearing on the results.

Line 145: Would you expect vergence movements during reading text on a screen? In the next sentence, were the variables mentioned (#fixations/min, fixation duration, % regressions) identified by the Tobii software or by an examiner from the recordings.

Figure 1: Did this person have strabismus? Why are the two eyes so misaligned? Why don't the eye movements look like the sorts of reading eye movements that have been published over many decades? Why all the vertical squggles? If this is an example of the best data obtained, what was the rest like? How could this have been analysed? The purpose of this figure is a true mystery.

Figure 2; Why is this so much better (and "normal") than Figure 1?

Section 2.3 Statistical analysis: This is an exceptionally thorough discussion of the statistical analyses. Could we also have some additional discussion about how the data being analysed were collected? There is no explanation about how the eye movement data were obtained and analysed.

Table 4: Where did smooth pursuit come from? It wasn't in the description of the NSUCO test earlier. And what do the scores refer to and how do the relate to actual measures of pursuit? 

Table 5: "Resolution time" suddenly appears. It's never been defined and I'm still not clear what it refers to.

Figure 3: If the graph is of fixations/min, shouldn't the Y-axis be a count, not a time? And what do the box, whiskers and X mean in the plot? Boxplots can be defined in many ways. 

Line 273: What sort of dysfunction? This lack of precision is a continual problem with the manuscript and makes interpreting the results difficult.

Lines 279-281: But if the scores are derived from multiple tests, can you say if the same elements contribute to the scores of the different groups? That's where actually recording the eye movements comes in. I'd have expected some sort of cross-validation between the subjective and objective tests. Again, lack of specificity hinders understanding.

Line 299: Finding that dyslexics can't read something well isn't exactly a novel finding.

Lines 306-312: It would be worth mentioning John Stein's work on visual pathways (especially the magnocellular pathway) in dyslexia; e.g., in Stein J. The magnocellular theory of developmental dyslexia. Reading and dyslexia: From basic functions to higher order cognition. 2018:103-34. It has direct relevance to the comments on ocular alignment in dyslexia.

Comments on the Quality of English Language

Please replace "oculomotricity" with a suitable English term, such as "ocular motility" or some more specific category of eye movement.

Author Response

Comment 1: Line 32: Define "oculomotricity." "Ocular motility" is a more common term in English.

Response 1: We appreciate the valuable feedback provided by the reviewers, which has significantly contributed to improving the quality of our manuscript.

Changed: Oculomotor function is the human ability to move the eyes naturally in a simple, coordinated, and smooth manner while maintaining a clear, fused, and fixed image on the central point of the retina [1]. We have changed the word oculomotricity to fixation [line 2], eye movements [lines 12], fixations during reading [line 15], "deleted" oculomotricity [line 31], fixations [line 92], eye movements [line 111], eye movements [136].

Comment 2: Lines 34-35: Other important types of eye movements are vergence and VOR. If the authors only refer to those relevant to reading, they should indicate this (although some researchers have linked vergence instability to dyslexia).

Response 2: Thank you for your insightful comments and suggestions. They have been very helpful in refining our study.

Changed: Other important types of eye movements are vergence and the vestibulo-ocular reflex (VOR). While this study primarily focuses on eye movements related to reading, it is important to note that some researchers have linked vergence instability to dyslexia [2].

Comment 3: Line 44: Regression is a specific term in reading. If only studying eye movements in reading, the introduction would be a good place to make this clear.

Response 3: I agree.

Added: In the study of eye movements, especially during reading, several technical terms are used. Regressions during reading are backward eye movements where the reader's gaze returns to previously read text. These movements are often involuntary and can occur when the reader needs to reprocess or clarify the content. Regressions are typically associated with difficulties in comprehension or decoding the text, reflecting the reader's need to revisit earlier words or sentences to enhance understanding. They are a natural part of the reading process but are more frequent in individuals with reading difficulties. According to recent research, skilled readers typically exhibit fewer than 15% of their eye movements as regressions per 100 words, indicating a higher efficiency in processing text compared to individuals with reading difficulties.

Added reference too: Inhoff AW, Kim A, Radach R. Regressions during Reading. Vision (Basel). 2019;3(3):35. doi:10.3390/vision3030035.

Coment 4: Line 71: Define "antisaccades" for the general reader.

Response 4: Thanks. Added: Antisaccades are voluntary eye movements where an individual must suppress a reflexive saccade towards a visual stimulus and instead look in the opposite direction. This task requires significant cognitive control, involving inhibition of the automatic response and the generation of a deliberate saccade in the opposite direction [7,8]. 

Added references 7 and 8: 

7. McDowell JE, Dyckman KA, Austin BP, Clementz BA. Neurophysiology and neuroanatomy of reflexive and volitional saccades: evidence from studies of humans. Brain Cogn. 2008;68(3):255-270. doi:10.1016/j.bandc.2008.08.016.

8. Lukasova K, Silva IP, Macedo EC. Impaired Oculomotor Behavior of Children with Developmental Dyslexia in Antisaccades and Predictive Saccades Tasks. Front Psychol. 2016;7:987. doi:10.3389/fpsyg.2016.00987.

Comment 5: Line 104: What sort of oculomotor anomalies? Strabismus, nystagmus or something else? This could have a bearing on the results.

Response 5: Deleted: "that may alter the visual system and eye movements." We have not found any literature with evidence indicating that medication for children with neurodevelopmental disorders can affect eye movements.

Added: and conditions such as amblyopia, strabismus and nystagmus.

Comment 6: Line 145: Would you expect vergence movements during reading text on a screen? In the next sentence, were the variables mentioned (#fixations/min, fixation duration, % regressions) identified by the Tobii software or by an examiner from the recordings.

Response 6: Thanks. Changed: The variables measured, which included the number of fixations per minute, average fixation duration, and percentage of regressions, were automatically identified and recorded by the Clinical Eye Tracking software.

Comment 7: Figure 1: Did this person have strabismus? Why are the two eyes so misaligned? Why don't the eye movements look like the sorts of reading eye movements that have been published over many decades? Why all the vertical squggles? If this is an example of the best data obtained, what was the rest like? How could this have been analysed? The purpose of this figure is a true mystery.

Response 7: 

We appreciate the reviewer's comments regarding Figure 1. In the original image, eye movements of each eye were represented in different colors (red for the right eye and green for the left eye) to illustrate the individual visual axes during reading. However, we understand that this representation may have caused confusion due to the apparent misalignment.

To address these concerns and improve the clarity of the data presentation, we have decided to replace Figure 1 with a new image. The new figure will show visual tracking while the subject performs the reading task. This new image more accurately and clearly represents the coordinated eye movements and provides a more representative display of the data obtained.

Comment 8: Figure 2; Why is this so much better (and "normal") than Figure 1?

Response 8: Thank you for your comment, I agree. As a change, we have selected an example in Figure 1 where the provided visual tracking is of better quality, eliminating most of the artifacts observed in the original figure. We believe that this modification will significantly improve the interpretation of the results and better align the figure with the typical eye movements described in the existing literature.

Comment 9: Section 2.3 Statistical analysis: This is an exceptionally thorough discussion of the statistical analyses. Could we also have some additional discussion about how the data being analysed were collected? There is no explanation about how the eye movement data were obtained and analysed.

Response 9: Thank you for your valuable feedback. We appreciate your suggestion to provide additional discussion on how the data were collected and analyzed.

*The sections on Description of Equipment and Software Used, Data Collection Procedure, and Measured Variables are explained and developed in the methodology section. However, we appreciate your review and will add the following: both the software and the eye tracker have been used in previous research, which assures the correct use of these tools.

Changed: The sections on Description of Equipment and Software Used, Data Collection Procedure, and Measured Variables are explained and developed in the methodology section. However, we appreciate your review and will add the following: both the software and the eye tracker have been used extensively in previous research, which assures the correct use and reliability of these tools. These prior studies have demonstrated the accuracy and validity of the Tobii Eye X eye tracker and the Clinical Eye Tracking software in capturing and analyzing eye movements. Consequently, the use of these well-validated instruments in our study enhances the credibility of our data collection process and supports the robustness of our findings.

Comments 10: Table 4: Where did smooth pursuit come from? It wasn't in the description of the NSUCO test earlier. And what do the scores refer to and how do the relate to actual measures of pursuit? 

Response 10: Thanks. I agree.

Smooth pursuit was not mentioned in the previous description of the NSUCO test and is not part of the methodology we described for the evaluation. Therefore, to maintain coherence and accuracy in our manuscript, we have decided to remove the smooth pursuit section from Table 4.

Comments 11: Table 5: "Resolution time" suddenly appears. It's never been defined and I'm still not clear what it refers to.

Response 11: I agree, added in line 185: resolution time, word per minute, and mean number of fixations per line. 

Comment 12: Figure 3: If the graph is of fixations/min, shouldn't the Y-axis be a count, not a time? And what do the box, whiskers and X mean in the plot? Boxplots can be defined in many ways. 

Response 12: Thank you for your valuable feedback. We appreciate your suggestion to clarify the labeling and interpretation of Figure 3.

Figure 3 presents a boxplot illustrating the number of fixations per minute (fixations/min) for the different groups studied. It is important to note that the Y-axis represents the count of fixations per minute rather than a unit of time. This adjustment will be made to correct the labeling discrepancy. The boxplot provides a summary of the distribution of fixation counts using five key statistics: the minimum, the first quartile (Q1), the median (Q2), the third quartile (Q3), and the maximum. The central box indicates the interquartile range (IQR), which encompasses the middle 50% of the data, while the whiskers extend to the smallest and largest values within 1.5 times the IQR from the quartiles. The 'X' mark within the box represents the mean fixation count. These elements collectively offer a visual representation of the central tendency and variability of fixation counts per minute among the groups. This detailed explanation ensures that readers can accurately interpret the data presented in Figure 3.

***Changed Figure 3 and 4.

Comment 13: Line 273: What sort of dysfunction? This lack of precision is a continual problem with the manuscript and makes interpreting the results difficult.

Response 13: Thank you for your valuable feedback. We appreciate your suggestion to provide more precision regarding the types of oculomotor dysfunctions.

Added:

Our aim was to investigate specifically the oculomotor pattern during the reading process to better understand the potential impact of various oculomotor dysfunctions in children with NDDs on reading performance. These dysfunctions include issues such as poor fixation stability, increased frequency of regressions (backward eye movements), and abnormal saccadic movements (rapid eye movements between fixation points). By examining these specific oculomotor abnormalities, we aim to elucidate how they contribute to difficulties in reading accuracy, speed, and comprehension among children with neurodevelopmental disorders.

Comment 14: Lines 279-281: But if the scores are derived from multiple tests, can you say if the same elements contribute to the scores of the different groups? That's where actually recording the eye movements comes in. I'd have expected some sort of cross-validation between the subjective and objective tests. Again, lack of specificity hinders understanding.

Response 14: We are grateful for your detailed observations, particularly regarding the methodology section. Your input has helped us clarify important aspects of our research process.

*Changed in materials and methods: 

To ensure the robustness of our findings, we performed a cross-validation by correlating the saccadic data obtained from the subjective NSUCO test with both the digitized NSUCO data and the Eye tracker data. Specifically, we correlated the variables of saccadic ability and precision. Patients were shown points of 0.5 cm diameter appearing on the screen every 1 second for 5 cycles.

To determine the relationship between subjective and objective oculomotor evaluations, we used Spearman's rank correlation coefficient, suitable due to the ordinal nature of NSUCO data and potential non-linearity. The ability evaluated by NSUCO corresponds to the complete saccades recorded by the eye tracker, and precision is equated with the number of hypometric saccades. We considered rho values of 0.00 < rho < 0.3 as weak, 0.3 < rho < 0.6 as moderate, and rho > 0.6 as strong.

**Results: To conduct the study using the objective test, a cross-validation was performed. For this purpose, we analyzed the correlation between the NSUCO test results and the eye tracker data. A significant moderate-to-high correlation was observed between the NSUCO test's assessment of saccadic ability and the number of complete saccades recorded by the eye tracker (r=0.702; p<0.001). Additionally, a significant correlation was found between the NSUCO-evaluated precision and the number of hypometric saccades detected by the eye tracker (r=0.686; p<0.001).

**Discussion: 

The findings of this study provide evidence of a significant correlation between the subjective assessments of the NSUCO test and the objective measurements obtained via an eye tracker. Specifically, the correlation between NSUCO-assessed saccadic ability and the number of complete saccades recorded by the eye tracker indicates that the NSUCO is a reliable predictor of complete saccades, a key indicator of oculomotor function. This suggests that as NSUCO-assessed ability increases, so does the number of complete saccades detected.

Furthermore, the significant correlation between NSUCO-assessed precision and the number of hypometric saccades detected by the eye tracker highlights the NSUCO's capacity to reflect saccadic precision, despite being a subjective measure. These results align with previous research that found relationships between subjective and objective oculomotor function measures, supporting the concurrent validity of both assessment methods. However, limitations include the non-representative sample, which may affect the generalizability of the results, and the fact that correlation does not imply causation, necessitating further studies to explore these relationships.

Comment 15: Line 299: Finding that dyslexics can't read something well isn't exactly a novel finding.

Response 15: In response to your suggestion, we have added:

**These results are consistent with those obtained by Tiadi et al. [22] and Raghuram et al. [23], which indicated that children with dyslexia took longer to read Card C compared to children without specific learning disorders. Similar findings were also reported by our research group in a previous study [14]. While it is well-established that children with dyslexia face challenges in reading tasks, our study provides novel insights into the specific oculomotor dysfunctions contributing to these difficulties. By using both subjective and objective measures, we were able to identify distinct patterns of poor fixation stability, increased regressions, and abnormal saccadic movements that underline the reading impairments in dyslexic children. These findings not only corroborate previous research but also enhance our understanding of the underlying mechanisms affecting reading performance in children with dyslexia.

Comment 16: Lines 306-312: It would be worth mentioning John Stein's work on visual pathways (especially the magnocellular pathway) in dyslexia; e.g., in Stein J. The magnocellular theory of developmental dyslexia. Reading and dyslexia: From basic functions to higher order cognition. 2018:103-34. It has direct relevance to the comments on ocular alignment in dyslexia.

Response 16: Based on your feedback, we have revised the relevant sections to provide a clearer explanation of our findings.

Changed: John Stein's magnocellular theory suggests that dyslexia is linked to dysfunctions in the magnocellular pathway, which is essential for processing motion and controlling eye movements. Stein argues that individuals with dyslexia have impairments in their visual magnocellular system, leading to challenges with visual motion sensitivity and binocular stability, both critical for effective reading. These deficits can cause the visual perceptual instability frequently observed in dyslexic individuals, where letters may appear to shift or blur. This instability can lead to common reading issues such as letter reversals and misordering. Stein further notes that genetic and environmental factors might affect the development of magnocellular neurons, connecting the neurobiological aspects to the reading difficulties encountered by dyslexic readers.

Round 2

Reviewer 1 Report

Comments and Suggestions for Authors

No comments

Author Response

I would like to express my gratitude for the time and effort dedicated to reviewing my manuscript. Your comments and suggestions have been extremely valuable and have significantly contributed to improving the quality of my work. Your detailed review and constructive feedback have been fundamental to the development and completion of this article.

Once again, thank

Reviewer 2 Report

Comments and Suggestions for Authors

The authors' responses have been thorough and have significantly improved the manuscript. There are a few remaining points to address.

Line 42: Tremor isn't OKN; it's actually called tremor. There is also drift and microsaccades in the category of fixational eye movements. See "Alexander RG, Martinez-Conde S. Fixational eye movements. Eye movement research: An introduction to its scientific foundations and applications. 2019:73-115." for a recent review. OKN is a ancient ocular response to large field visual motion, present even in some invertebrates.

Line 152: What are "complete saccades"? This isn't a familiar term.

Line 195: Please define "resolution time".

Legend, Figure 1: The eye movement trace is now blue; there's no red or green.

Author Response

Comentario 1: Línea 42: El temblor no es OKN; en realidad se llama temblor. También hay deriva y microsacadas en la categoría de movimientos oculares fijacionales. Consulte "Alexander RG, Martinez-Conde S. Movimientos oculares fijacionales. Investigación del movimiento ocular: una introducción a sus fundamentos científicos y aplicaciones. 2019:73-115." para una revisión reciente. OKN es una antigua respuesta ocular al movimiento visual de campo amplio, presente incluso en algunos invertebrados.

Respuesta 1: Gracias por ayudarnos a mejorar la precisión y claridad de nuestro manuscrito. 

Modificado: Las fijaciones son la capacidad voluntaria de mantener la mirada sobre un estímulo visual específico. Sin embargo, los ojos no permanecen completamente inmóviles durante la fijación, sino que presentan pequeños temblores, desviaciones y microsacadas totalmente involuntarias e independientes para cada ojo, con amplitudes inferiores a 1°. Estos movimientos oculares de fijación son los encargados de mantener la imagen en la fóvea para evitar que la imagen obtenida aparezca borrosa, ya que al realizar estos pequeños movimientos se evita la saturación de los fotorreceptores. 

Se agregaron  Alexander RG, Macknik SL, Martinez-Conde S. Microsaccades in Applied Environments: Aplicaciones reales de las mediciones del movimiento ocular por fijación. J Eye Mov Res. 2020 ;12(6):10.16910/jemr.12.6.15. doi:10.16910/jemr.12.6.15

Comentario 2: Línea 152: ¿Qué son los "movimientos sacádicos completos"? No es un término muy conocido.

Respuesta 2: Agradecemos el comentario del revisor. Esperamos que esta definición aclare ambos conceptos para una mejor lectura.

Modificado: La capacidad evaluada por NSUCO corresponde a los movimientos sacádicos completos registrados por el rastreador ocular, lo que se refiere a la capacidad de realizar un movimiento ocular completo hacia el estímulo de fijación. Esto es equivalente a la variable de capacidad en NSUCO. La precisión, según la evaluación de NSUCO, se equipara con el número de movimientos sacádicos hipométricos registrados por el rastreador ocular, lo que significa que los movimientos oculares no alcanzan el estímulo objetivo. 

Comentario 3: Línea 195: Defina "tiempo de resolución".

Respuesta 3: Agradecemos el comentario del revisor. El tiempo de resolución se refiere a la duración total que requiere un sujeto para leer un texto determinado desde el principio hasta el final. Esto abarca todo el período desde la fijación inicial en la primera palabra hasta la fijación final en la última palabra. Esta definición se ha añadido al manuscrito para mayor claridad.

Añadido: El tiempo de resolución se refiere a la duración total necesaria para que un sujeto lea un texto determinado desde el principio hasta el final, abarcando todo el período desde la fijación inicial en la primera palabra hasta la fijación final en la última palabra.

Comentario 4: Leyenda, Figura 1: El rastro del movimiento ocular ahora es azul; no hay rojo ni verde.

Respuesta: Agradecemos la atención del revisor a este detalle. El trazo del movimiento ocular ahora está representado por una línea azul. Este cambio se realizó para evitar confusiones con respecto a los movimientos de fijación de cada ojo por separado, lo que podría haber sugerido incorrectamente la presencia de estrabismo. Esta aclaración es importante ya que ninguno de los sujetos incluidos en el estudio tenía estrabismo, ya que era un criterio de exclusión. Por lo tanto, optamos por incluir una figura que represente la fijación del sujeto durante la lectura.

Añadido: (Leyenda fig 1.) ** Texto de lectura elegido para evaluar movimientos sacádicos cortos. La línea azul representa el movimiento de fijación del sujeto a lo largo de la lectura del texto. 
